# RoBERT: Low-Cost Bi-Directional Sequence Model for Flexible Robot Behavior Control

## Abstract

Requirement of human involvement for data collection or system design has always been a major challenge for building robot control policy. In this paper, we present **Ro**bot-**BERT** (RoBERT), a method to build general robot control policy for complex behaviors with *least* human effort. Starting from unsupervisedly-collected dataset, RoBERT has no requirements of human labels, high-quality behavior dataset or accurate information of system model, in contrast to most other methods for building general robot agent. RoBERT is further unsupervisedly trained via *Masked Action-Inverse-Inference* (MAII), a method inspired by *Masked Language Modeling* (MLM) in BERT-like language models and has potential to enable *zero-shot*, *multi-task*, *keyframe-based* robot control with little architectural change and user-friendly interface. In our empirical study, RoBERT is successfully applied on various types of robots in simulated environment and could generate stable and flexible behaviors to fulfill complex commands.

## 1 Introduction

Recent years have witnessed a surge of research interest to apply Transformer-based (Vaswani et al., 2017) large sequence model (a.k.a Large Language Model, LLM) into multiple machine learning related fields, such as Computer Vision (CV) (Dosovitskiy et al., 2021), Natural Language Processing (NLP) (Devlin et al., 2019; He et al., 2020; Brown et al., 2020) and Decision Making (Reed et al., 2022; Lee et al., 2022; Chen et al., 2021; Janner et al., 2021).

Among them, using pretrained large sequence model, combined with algorithm in Reinforcement Learning (RL) or Imitation Learning (IL) to build general robotic agent is of much interest and importance (Brohan et al., 2023b;a; Huang et al., 2023), as conventional model-based method usually struggles in complex robotic system (Lyu & Cheah, 2020; Cui et al., 2021).

However, existing works on this field either rely on large amount of human-labelled multi-modal data (Brohan et al., 2023b;a; Huang et al., 2023) or near-optimal behavior dataset (Reed et al., 2022; Brohan et al., 2023a) that are expensive to collect. In our insight, the need of such human effort would hinder the development of large sequence model for robot control, owing to the lack of reusable data and expert resources in community of Robotics.

Inspired by recent advancements in *Reward-free* and *Unsupervised RL* (Jin et al., 2020; Tarbouriech et al., 2020; Chen et al., 2022; Yarats et al., 2021; Liu & Abbeel, 2021b; Laskin et al., 2021; Burda et al., 2018b), we propose a new way to both collect dataset and train models with little human effort. It means: 1) Dataset is collected by maximizing *intrinsic rewards* without any *extrinsic reward* (and *reward engineering*). 2) Models are then trained directly and purely on this collected dataset following methods inspired by Behavioral Cloning (BC) and Curriculum Learning (CL) (Wang et al., 2022). 3) During deployment, user-supplied commands *extract* behaviors needed, enabling *goals-conditioned* or even *keyframe-based open-loop* control *zero-shotly*.

What's more, a method trained in this way is naturally *multi-task*. Our insight is that every specified reward function would introduce a *bias* towards what behaviors an agent should generate. By encouraging an agent to perform task in one direction, we are discouraging it from performing task in opposite direction. For example, by tweaking reward function to train a robot to teach dance smoothly, we're restricting its ability to serve as a counter-example for showing how un-smooth dances look like.

In summary, the main contributions of this paper are as follows:

1. We present RoBERT, a method to both collect dataset and train sequence models at low cost. To the best of our knowledge, our method requires *the least* amount of human labelling, expert behavior dataset or system knowledge, compared to other state-of-the-art methods for general robot control/decision-making.

2. To the best of our knowledge, we are the first to show that with recent advancements in unsupervise RL and computing ability, RoBERT could be readily instantiated on rather complex robot models (Universal Robot 5e robot arm and Shadow E3M5 robot hand) with dataset of only millions of transitions.

3. We perform extensive experiments and analysis to demonstrate our instantiated RoBERT could successfully generate flexible and stable actions on various robot models and are robust against hyperparameters and model architecture. We also plan to release dataset and code used in this work as a testbed for future researches.

## 2 RELATED WORK

### 2.1 SEQUENCE MODELING IN DECISION MAKING

Applying sequence model into field of decision making has been a popular research direction since the recent success of LLM in NLP area. Chen et al. (2021) and Janner et al. (2021) firstly introduce Transformer into model-free and model-based RL method respectively and show promising results in many RL tasks. Following their work, Furuta et al. (2021) further investigates various ways to integrate Transformer into various RL/IL setting and Kim et al. (2023) integrates Transformer into Preference-based RL (PbRL). In IL, Shafiullah et al. (2022) proposes to use Transformer for multi-modal behavior modelling and their idea is evaluated into our method. Besides, some works also focus on Representation Learning by masking on bi-directional Transformer encoder-decoder, as in Liu et al. (2022); Wu et al. (2023). Our work mainly differs from theirs as ours focuses on instantiating a control policy from dataset collected purely from intrinsic rewards and show that decent performance coule be achieved with only millions of transitions.

### 2.2 REWARD-FREE AND UNSUPERVISE RL

Reward-free RL proposes to study how to perform exploration without knowing task/reward function *a prior* and collects dataset which could enable near-optimal policy learning for *any* task specified afterwards (Jin et al., 2020; Tarbouriech et al., 2020; Chen et al., 2022). Many works in this direction focus on theoretical results. More practically, Unsupervise RL employs various *intrinsic reward* (prediction error, distribution entropy, etc.) to maximize statistical properties of trajectories collected, in order to collect diverse behavior dataset or learn task policy efficiently. Burda et al. (2018a); Laskin et al. (2021); Yarats et al. (2022); Liu & Abbeel (2021b); Lobel et al. (2023); Liu & Abbeel (2021a); Yarats et al. (2021); Eysenbach et al. (2018) are all works in this line that differ mainly on optimization objectives selected. In this work, we employ various unsupervise RL methods and mix datasets their collected, as a resource of diverse behaviors (more in Section 3.2).

### 2.3 ROBOTICS

Applying Neural Network (NN) and RL/IL into field of robot control has gained great achievements recently. For example, on locomotion task, quadruped robot has been successfully trained to move on various terrains with ability of fast-adaptation or self-recovery (Nahrendra et al., 2023; Choi et al., 2023; Miki et al., 2022; Yang et al., 2020). In simulation environment, Peng et al. (2018); Tessler et al. (2023); Peng et al. (2022); Chen et al. (2023) are also works commanding robot character performing complex sequential behaviors for animation generation. Recently, works combining LLMs and Robotics have also emerged. In Brohan et al. (2023b;a), VLA (Vision-Language-Action) Model based control policy could enable robot arm to follow human instruction and complete various tasks. Also, Huang et al. (2023) tries to employ LLM as high-level planner and is also evaluated on robot arm. Different from theirs, ours investigates how to use Transformer model for Robotics in a low-cost and accessible way.

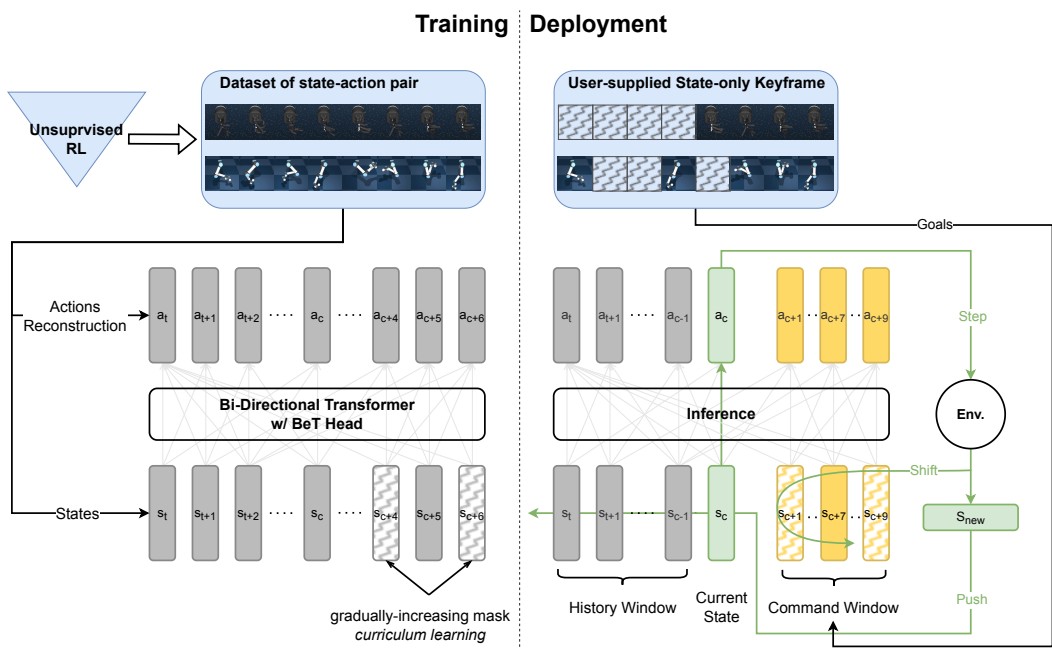

Figure 1: Framework of RoBERT. States with zigzagged line represents masked ones.

## 3 RoBERT

The framework of RoBERT is given in Figure 1. In the training phase, trajectories of state-action pair are feed into Bi-Directional Transformer with gradually-increasing (CL-based) masking applied in the same 'command window' of deployment. During deployment, the model is queried with input concatenated from 'history window', current state and 'command window', then the output action corresponding to current state is rollouted in environment. After each rollout, new state would be back-pushed into history-current window while command window would be left-shifted.

### 3.1 PRELIMINIARIES

Consider a rewardless Markov Decision Process (MDP), $\mathcal{M} = (\mathcal{S}, \mathcal{A}, \mathcal{T}, \mathcal{S}_0)$ where $\mathcal{S}$ is set of all states, $\mathcal{A}$ is set of all actions, $\mathcal{T}$ is transition dynamics $\mathcal{T}(\mathcal{S}'|\mathcal{S}, \mathcal{A}) \to \mathcal{R}$ that defines probability distribution of next states given previous state and action taken, and $\mathcal{S}_0$ is distribution of initial states. Objective of Imitation Learning is to find a policy $\pi$ that imitates behaviors present in a dataset $D = \{(s, a)\}$:

$$\pi = \arg\min_{\pi} \mathbb{E}_{s, a \sim D} \mathcal{L}(\pi(s), a)$$

where $\mathcal{L}$ is some loss function (e.g. Mean Square Loss (MSE) when $\pi$ is deterministic or Softmax between loglikelihood when $\pi$ is stochastic and action set is finite). Finally, we introduce *inverse dynamics function* $\mathcal{I} = \mathcal{I}(s_t, ..., s_{t+h}) \to (a_t, ..., a_{t+h})$ that outputs action sequence corresponding to input state sequence.

### 3.2 TRAVERSAL OF STATE SEQUENCE SPACE

Consider a dataset of trajectory of length $h$, $D = \{(s_t, a_t, s_{t+1}, a_{t+1}, ..., s_{t+h}, a_{t+h})\}_{t=1}^{N}$, note $s_t$ isn't necessarily initial state $s_0$ and could be any state of a trajectory, so long as length of its following steps exceeding $h$. If we obtain such a dataset that distribution of its state sequence ($D_s = \{(s_t, s_{t+1}, ..., s_{t+h})\}_{t=1}^{N}$) is diverse enough, i.e.:

$$\forall s_q^h = (s_q, s_{q+1}, ..., s_{q+h}) \in \{\text{all possible state seq.}\} \Longrightarrow$$

$$\exists s_i^h = (s_i, s_{i+1}, ..., s_{i+h}) \in D_s \quad s.t. \quad \mathcal{L}(s_i^h, s_q^h) \leq \theta$$

where $\mathcal{L}$ is some loss function, $\theta \in \mathcal{R}^+$ is a measure of diversity of $D_s$ and while $\theta$ approaching 0, $D_s$ is more diverse in state sequence space.

Then, if we have a near-optimal policy $\pi$:

$$\pi \quad s.t. \quad \forall s_i^h \in D_s, \ \mathcal{L}(a_i^h, \pi(s_i^h)) \le \epsilon$$

where $\mathcal{L}$ is loss function, $a_i^h = (a_i, a_{i+1}, ..., a_{i+h})$ is series of actions corresponding to $s_i^h$, $\pi(s_i^h) = (\pi(s_i), \pi(s_{i+1}), ..., \pi(s_{i+h}))$ is action sequence output by $\pi$, $\epsilon \in \mathcal{R}^+$ is a measure of optimality of $\pi$ and while $\epsilon$ approaching 0, $\pi$ is more optimal.

The insight here is the data-driven learning process above could yield a decent function approximator for inverse dynamics, i.e.:

$$\text{for } \pi = \arg\min_\pi \ \mathbb{E}_{s^h \in D_s} \ [\mathcal{L}(a^h, \pi(s^h))] \ \text{ and } \ \forall s_q^h \in \{\text{all possible state seq.}\}$$
$$\text{we have } \ \mathcal{L}(\mathcal{I}(s_q^h), \pi(s_q^h)) \le \delta$$

where $\mathcal{L}$ is loss function and $\delta \in R^+$ is a bounded value.

Therefore, with a near-optimal $\pi$ on a decently diverse dataset $D$, we could perform multi-goals conditioned control by concatenating history window, current state and remaining goals to query $\pi$ for recovered action series.

### 3.3 MASKED ACTION-INVERSE-INFERENCE (MAII)

Despite $\pi$'s ability to serve as a multi-goals conditioned policy, specifying series of states as goals is still a tedious process and not user-friendly. Therefore, inspired by Masked Language Modeling (MLM) in BERT (Devlin et al., 2019) and Curriculum Learning (CL), we design MAII, a way to introduce masking into training of RoBERT for keyframes control. In details, user-supplied commands could contain mask token same as the one applied during training phase, indicating 'no goal specified' on these keyframes.

An intuitive idea here is as masks increasing, determinism of inverse dynamics would decrease and our setting would become a multi-modality Imitation Learning, as studied in Shafiullah et al. (2022). Hence, we also test technique in Shafiullah et al. (2022): output index of kmeans cluster and offset to it, then combine them back to continuous actions (please refer to Shafiullah et al. (2022) for details).

### 3.4 KEYFRAME-BASED OPEN-LOOP CONTROL

After training, model is fixed and tested as a zero-shot policy. As Figure 1 shows, model keeps a history window that tracks last few states and a future/command window specifying remaining commands to achieve. With insertion of current state in between, this composes a state sequence same as training time.

We choose only position values of joints as states in this work because 1) they are easy to interpret, specify and tweak. 2) a well-trained model shall have ability to infer velocity and other higher-order derivatives automatically given a series of position observations. We also choose open-loop control since close-loop control conflicts with time-aware keyframe control.

## 4 MAIN EXPERIMENTS

### 4.1 GENERAL SETTING

We briefly describe general setting of our main experiments in this section. Then we present our main results on 3 distinct robots: Pointmass, Universal Robot 5E arm (UR5E) and Shadow Hand robot hand (Shadow Hand) as shown in Figure 2. For more details of our experiment settings, please refer to Appendix A.1 and our code.

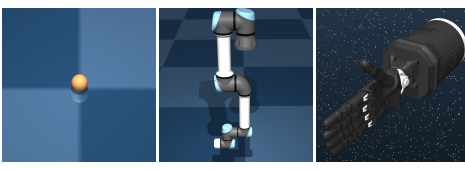

Figure 2: Environments used: (from left to right) Pointmass, UR5E and Shadow Hand

### 4.1.1 DATASET COLLECTION

As explained in Section 2.2, we employ existing unsupervise RL methods adopted from URLB (Laskin et al., 2021) codebase and combine dataset collected by them . We choose dataset size as a *rule of thumb* and tend to collect excessive amount as shrinking dataset is cheaper than re-collecting.

Table 1: Outline of environment, dataset and model settings

| Env. | State Dim. | Action Dim. | #. Transitions[a] Collected | #. Model Params. |
|---|---|---|---|---|
| Pointmass | 2 | 2 | 306M[b] | 0.11M $\sim$ 50.4M |
| UR5E | 5 | 5 | 41.9M | 42.6M |
| Shadow Hand | 22 | 18 | 131.3M | 42.73M |

[a] Transitions here refer to one step of state-action.
[b] We deliberately collect excessive amount of data for testing scaling effects, as shown in Section 4.2.1.

### 4.1.2 TRAINING

During training, we firstly split collected trajectory dataset into training and test set (95%-5% ratio). From test set, we randomly extract dozens of trajectories and combine them with several manual-designed tasks (for instance, *stay in the origin*) to form the final test set. Then we train our model on training set only (default hyperparameters in Appendix A.2) and calculate *Precision* (Mean Absolute Error (MAE)) between positions achieved and positions desired[1] as a metric of performance and pick best model based on it. Also note we calculate precision over strictly aligned timeframes and don't allow time-warping.

### 4.1.3 EVALUATION - TASK FULFILLMENT

We design experiments of task fulfillment to test how well RoBERT could generate behaviors to fulfill given commands/tasks in a keyframe-based manner. All experiments are conducted in a single episode without reset. All tasks span over 10 seconds while some exceeding 20 seconds. It's also worth noting that all tasks are *unknown* during data collection and model training phase, hence illustrating general and multi-task ability of RoBERT.

For each robot, we present visual results (2D coordinates plotting for pointmass or video clips for others) and diagram of *Precision-Time* to illustrate time-axis stability of RoBERT. We highly recommend readers to refer to videos in supplement material (Appendix A.4 and uploaded file) for a best view.

### 4.2 POINTMASS

A pointmass is controlled by 2 slide joints on an infinitely large plane. State space is 2 cartesian coordinates (*xy*) of it. We perform direct force control in this environment.

### 4.2.1 SCALING EFFECTS

We firstly investigate relationship between size of collected dataset and performance of trained model in order to guide our following data collection and training on more complex environments (Details in Appendix A.1.1).

As observed in Figure 3: 1) As dataset size reduces, models with various size and learning rate obtain worse performance. 2) In pointmass environment, effects of saturation is observed as enlarging dataset

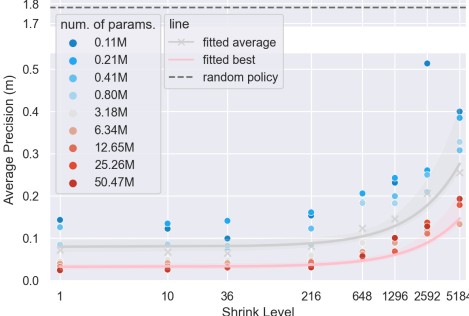

Figure 3: Scaling effects of Pointmass: *shrink* means by which ratio the original dataset is reduced and *Average Precision* is average of lowest precision observed in their training process. Lightgrey line is fitted for average values in a shrink while pink line is fitted for best value. Translucent bands are 95% confidence interval.

---

[1]Masked keyframes are excluded from precision calculation but included in loss calculation.

brings only insignificant performance increase. 3) Larger model usually obtains better results, compared to small-sized one.

Because we use precision as the metric, a relatively small difference in precision may already cause a severe performance degradation. To illustrate this, we choose the best model in shrink 5184 and test its performance on task fulfillment as in Section 4.2.2. Recorded videos are given in supplement materials and also aligns with our expectation that performance suffers from significant degradation.

### 4.2.2 PERFORMANCE ON TASK FULFILLMENT

In this section, we instruct RoBERT to fulfill multiple complex tasks. We choose the best model in shrink 10 for task fulfillments and for each task, we draw plots of trajectory on 2D coordinate and their tracking loss. We also note that commands given here may itself be *physically impossible* such as the sharp turn in *triangle* task. Results are shown in Figure 4.

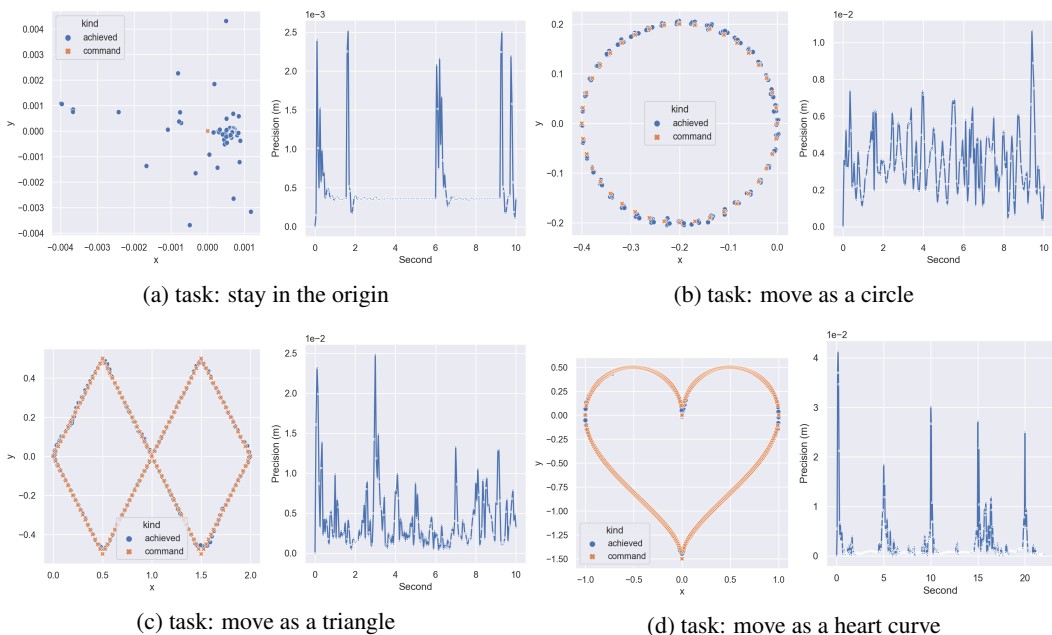

(a) task: stay in the origin          (b) task: move as a circle

(c) task: move as a triangle          (d) task: move as a heart curve

Figure 4: Results of RoBERT performing multi-task in pointmass (best viewed in videos). In coordinate plot, circle point is achieved position while crossmark point is commanded point.

### 4.3 ROBOT ARM - UR5E

An UR5E robot arm is controlled by 5 hinge joints with its base fixed in the air and normal gravity enabled. State space is 5 radian positions of each joint. Action space is torque directly applied to each joint.

As illustrated in Figure 3, models with different size and hyperparameters tend to have similar performance when dataset is decently large and diverse. Thus we don't perform hyperparameter-tuning on UR5E and ShadowHand. This also aligns with our insight that with recent progress in unsupervise RL we could already obtain a general robot control policy readily.

### 4.3.1 PERFORMANCE ON TASK FULFILLMENT

In this section, we present results of commanding RoBERT-UR5E to perform multiple tasks. The first task is to perform a 'reacher'-like task where robot arm needs to rotate while keeping *elbow joint* fixed to 90 degrees. Second one is to 'swing' like a clock tick while moving only *shoulder lift joint*. Videos clips are shown in Figure 5.

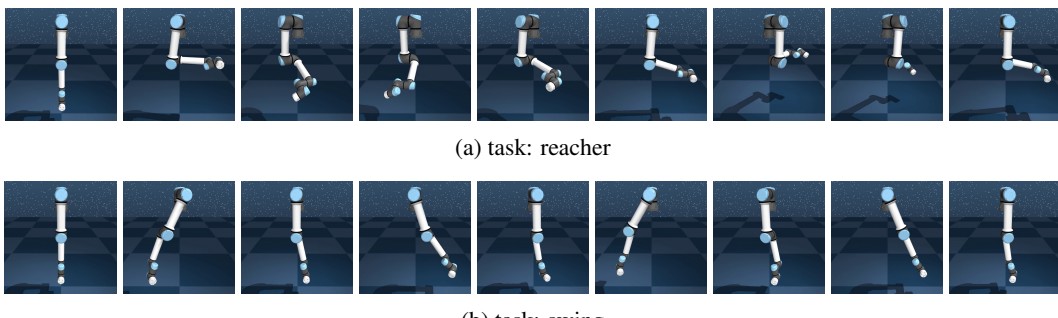

(a) task: reacher

(b) task: swing

Figure 5: Video clips of RoBERT controlling UR5E (best viewed in videos).

### 4.3.2 PERFORMANCE ON KEYFRAME-BASED CONTROL

In this section, we test RoBERT-UR5E's ability to fulfill commands when masks enabled.

The first command is to let robot arm 'hang' its elbow joint in left and right side alternatively, while keeping other parts fixed. All transition frames that happen in between of these 2 fixed posture are masked. The second command is commanding robot arm to reach front-left and back-left (named 'front_back') alternatively while masking transition frames.

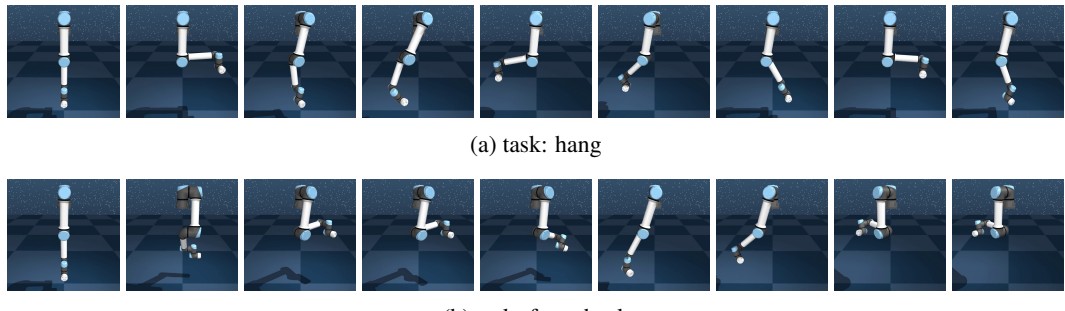

(a) task: hang

(b) task: front_back

Figure 6: Video clips of RoBERT controlling UR5E (best viewed in videos).

### 4.4 ROBOT HAND - SHADOW HAND

A Shadow Hand with 22 hinge joints are controlled by 18 actuators. There are 14 actuators applying torques directly on 14 hinge joints while 4 actuator applying control signal on 4 tendons, each linking 2 joints. State space is radian positions of 22 joints. Action space is control signal applied in each actuator. Gravity is removed in this environment.

### 4.4.1 PERFORMANCE ON KEYFRAME-BASED CONTROL

We test how well RoBERT could control Shadow Hand performing various tasks in *keyframe-based* manner. 2 tasks in Figure 7 are 1) command hand to pose a 'six' posture. 2) command hand to pose a 'thumb' posture, respectively. In these tasks, transition frames that leading hand from initial posture to desired ones are masked.

### 4.4.2 PERFORMANCE ON SEQUENTIAL KEYFRAME-BASED CONTROL

Due to the nature of RoBERT, it's easy to command for sequential and dynamic behaviors. In this section, we demonstrate 2 commands that test this feature of RoBERT. The first one is to perform 'fire a handgun' while the second one is playing 'rock-scissor-paper' game, as shown in Figure 8. In these tasks, transition frames leading hand from initial posture to desired ones are masked.

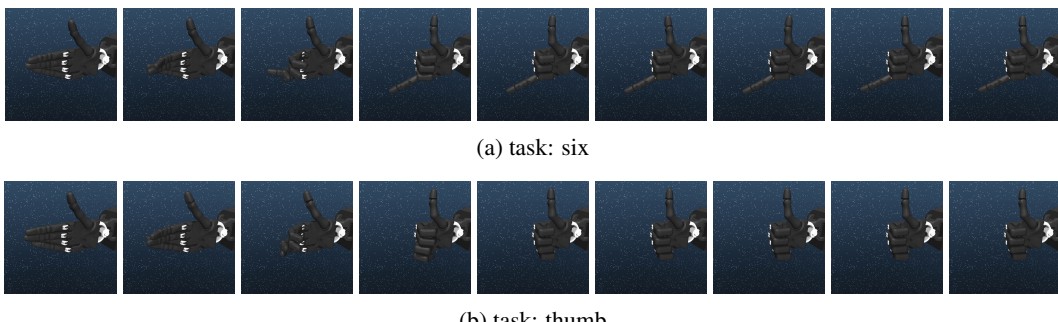

(a) task: six

(b) task: thumb

Figure 7: Video clips of RoBERT controlling ShadowHand (best viewed in videos).

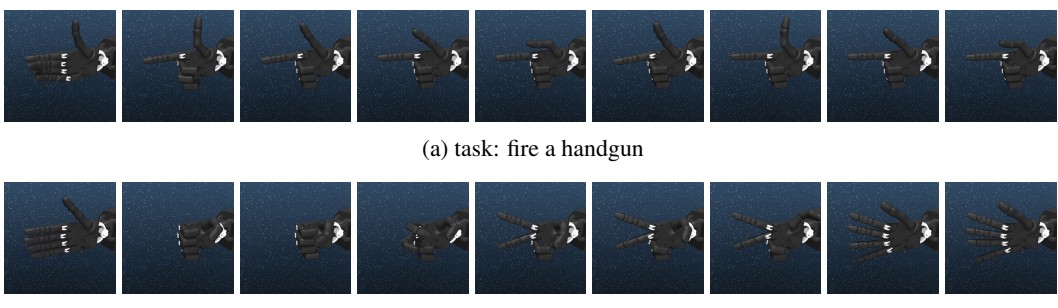

(a) task: fire a handgun

(b) task: rock-scissor-paper

Figure 8: Video clips of complex behaviors of RoBERT on Shadow Hand (best viewed in video).

## 4.5 ANALYSIS

In this section, we investigate how several aforementioned components affect model performance: 1) How robust is RoBERT against ill-collected dataset? 2) How does number of action bins affect model performance? 3) Is CL-based masking improving model performance?

To answer these questions, we choose ShadowHand and evaluate its performance over 2 sets of tasks extracted from test set in training process: 1) 'no-mask' tasks where tasks contain no masking. 2) 'high-mask' tasks where tasks have significant potion (50%) of masking [2]. All experiment settings are run in 3 seeds. Results are shown in Figure 9.

### 4.5.1 DATASET CORRUPTION

We firstly test how an ill-collected dataset would affect performance of RoBERT. To do so, we deliberately corrupt collected dataset by replacing certain proportion of its trajectories by a randomly picked trajectory. Then we perform normal training over these corrupted dataset and obtain results in Figure 9a.

From Figure 9a, we find: 1) On both no masking and high masking tasks, low corruption ratio doesn't significantly hinder performance of RoBERT. 2) For high corruption ratio ($\geq 30\%$), the decrease of RoBERT's performance is still in the same magnitude compared to the original one. Recorded video of RoBERT trained on corruption ratio 45% also shows no significant performance degradation, reflecting RoBERT's robustness towards a skewed dataset.

---

[2]We also add task 'rock-scissor-paper-high-mask' and 'fire-handgun-high-mask' where mask ratio is deliberately increased for better evaluation.

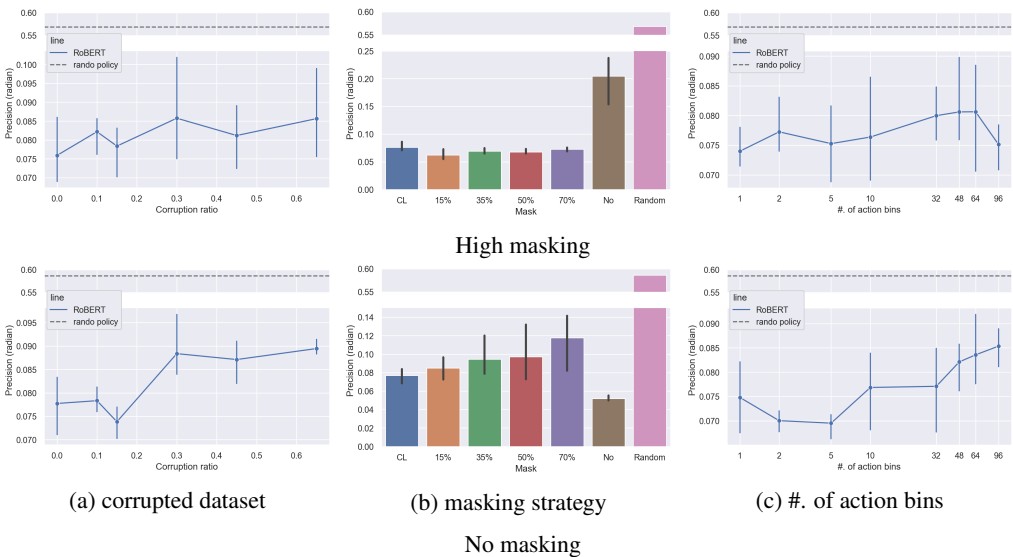

Figure 9: Analysis for different settings of RoBERT. Errorbar represents 95% confidence interval.

### 4.5.2 MASKING STRATEGY

In this section, we conduct experiment to test how different masking strategy affect model training. We select 6 masking strategy: 1. CL-based masking used in main results. ('CL'). 2. fixed masking ratio (15%, 35%, 50%, 70%) 3. no masking at all ('No').

Results are shown in Figure 9b. What we observe is: 1. No masking, as expected, leads to worst performance on high masking tasks but best performance on no-masking tasks. 2. static masking of low ratio could already achieve decent performance compared to CL-based one.

We also record videos for no masking and 15% static masking ratio. Results suggest: 1) no masking, 15% masking and default CL masking obtain similar precision over unmasked keyframes. 2) for masked frames, 'no masking' results in worst precision while 15% static masking performs best and default setting in between.

### 4.5.3 NUMBER (#) OF ACTION BINS

As in Section 3.3, we also tried the method in Shafiullah et al. (2022) to tackle multi-modality situation introduced by keyframe-based control. Herein, we experiment over various choices of number of action bins (1, 2, 5, 10 (default), 32, 48, 64 and 96) and obtain the result in Figure 9c.

We find: 1) Large number of action bins would increase performance for high-masking tasks but hinder performance on no-masking tasks. 2) Surprisingly, no multi-modality (action bins = 1) already achieves decent performance, compared to multi-modal version.

We also record videos for action bins 1 and 2, and find: 1) Only 1 action bin leads to most stable behavior generation without serious performance degradation 2) Using 2 action bins slightly introduces 'shaking' in action generation but still smaller than default. This shows that setting action bins to small values would generally enable control both stably and precisely.

## 5 SUMMARY

In this paper, we present RoBERT, a low-cost method to obtain bi-directional sequence models for *zero-shot*, *multi-task* and *keyframe-based* robot control. Extensive experiments and analysis are conducted, to demonstrate that owing to recent progress in unsupervise RL and computing ability, RoBERT could be obtained with minimum human involvement and serve as a robust foundation model for further improvements. We believe in the field of LLM-based Robotics, the accessibility of RoBERT would enable more researches and participations that benefits the whole community.

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

## A  APPENDIX

### A.1  DETAILS OF MAIN EXPERIMENTS

All experiments are conducted in Mujoco simulator (Todorov et al., 2012). Model specifications of UR5E and Shadow are adopted from mujoco_menagerie (Contributors, 2022). We use a control rate of 20Hz for all envs. In all envs., model output is directly applied as action signal (force or torque) without any further processing.

For all experiments, we use a length of history window of 30 frames (1.5s) and length of command window of 20 frames (1s).

To collect dataset, we train unsupervise RL algorithm (namely, RND (Burda et al., 2018b), APT (Liu & Abbeel, 2021b), APS (Liu & Abbeel, 2021a), ICM (Pathak et al., 2017) and Proto (Yarats et al., 2021)) for some large total steps and record network snapshot after a small step. Then we use these snapshots to rollout in environment, collecting trajectories of state-actions as dataset. At last, the datasets are merged together to form the final dataset used in each environment. Usually we tend to set the total step as a large number *rule-of-thumb*-ly and it works quite well. But for UR5E where unsupervise RL algorithm tends to collect extreme behaviors, we stop data collecting phases relatively early and this is the only tuning we performed.

For Pointmass and ShadowHand, we extract 18 (6 unmasked + 6 low-masked + 6 high-masked) trajectories from test set. For UR5E, we extract 90 (30 unmasked + 30 low-masked + 30 high-masked) trajectories. Then these trajectories are merged with manual-designed tasks (15, 4, 18 for Pointmass, UR5E and ShadowHand, respectively).

For test tasks used during training phase, we restrict their length to be 110 frames (30 for history window + 80 for commands length) as a constraint of time and computing resources.

We tune learning rate with a linear warm up and cosine annealing as in Brown et al. (2020).

The CL-based masking starts from 0 and gradually increases to 'max' (only 1 frame is unmasked).

### A.1.1 DETAILS OF SCALING EFFECTS

To experiment over scaling effects on pointmass, we first collect a large dataset (304M transitions) then shrink it to smaller size by sub-selecting trajectories with a fixed step and adjust learning rate and batch size slightly to fit for small-sized dataset. We run 3 seeds for each setting and random policy then report mean of their performance in Figure 3.

### A.2 DEFAULT HYPERPARAMETERS

We list default settings used for model training in Pointmass, UR5E and ShadowHand environments in Table 2.

Table 2: Default hyperparameters used in model training

| Hyperparameter | Value | Remarks |
|---|---|---|
| Learning Rate | 7e-5 | learning rate without scheduling |
| Total Training Frames | 5.12M (Pointmass) 6.144M (UR5E) 7.68M (ShadowHan) | |
| Learning Rate Warm-up | 10% | linear warm-up from 0.01% to 100% |
| Cosine Annealing begins | 35% | cosine annealing from 100% to 0.1% |
| Cosines Annealing ends | 85% | cosine annealing from 100% to 0.1% |
| Batch Size | 512 | |
| Hidden Dim. of Transformer | 768 | |
| Num. Layers of Transformer | 6 | |
| Num. Head of Transformer | 6 | |
| Num. of Action bins | 10 | |
| Masking begins | 2% | linear increasing mask ratio |
| Masking ends | 62% | linear increasing mask ratio |

### A.3 DETAILS OF ANALYSIS EXPERIMENTS

We use the same setting of main experiments for analysis experiments.

### A.4 LIST OF CONTENTS IN SUPPLEMENT MATERIALS

We have uploaded supplement materials and list contents of it here as a quick reference:

- code: This is training/evaluating code for RoBERT as used to produce results in this paper.
- videos: all video results:
  - main_result: these are main results for Pointmass, UR5E and Shadow Hand environments.
  - pointmass_shrink_5184: these are task fulfillment results for RoBERT trained in shrinked pointmass dataset.
  - analysis: results for Analysis section:
    * shadow_default: This is model trained with default setting on tasks of analysis section. This is mainly for reference purpose.
    * shadow_corrupt_0.45: This is model trained with corrupted dataset.
    * shadow_mask_15: This is results of model trained with static 15% masking.
    * shadow_mask_no: This is results of model trained with no masking applied.
    * shaodw_num_center_1: This is results of model trained with only 1 action bin.
    * shdow_num_center_2: This is results of model trained with 2 action bins.

