# OpenReview forum: "RoBERT: Low-Cost Bi-Directional Sequence Model for Flexible Robot Behavior Control"
_ICLR.cc/2024/Conference — ICLR 2024 Conference Withdrawn Submission_

### Official Review · Reviewer_piq4 · 2023-10-25

**Soundness:** 2 fair
**Presentation:** 1 poor
**Contribution:** 2 fair
**Rating:** 3
**Confidence:** 4

**Summary:**

This paper introduces Robot-BERT (RoBERT), a method for building a general robot control policy for complex behaviors with minimal human involvement. Unlike traditional methods, RoBERT does not require human labels, high-quality behavior datasets, or accurate system model information. It is trained through Masked Action-Inverse-Inference (MAII), similar to Masked Language Modeling in BERT-like language models, making it suitable for zero-shot, multi-task, and keyframe-based robot control with minor architectural changes. Empirical studies demonstrate RoBERT's success in generating stable and flexible behaviors for various robot types in simulated environments.

**Strengths:**

- Interesting application of masked sequence models into robot trajectory modeling.
- Thorough experimentation on different types of robots and tasks.

**Weaknesses:**

The presentation and writing of this paper makes it very difficult to assess the quality of this work.  There are incoherent sentences (eg: `By encouraging an agent to perform task in one direction, we are discouraging it from performing task in opposite direction. For example, by tweaking reward function to train a robot to teach dance smoothly, we’re restricting its ability to serve as a counter-example for showing how un-smooth dances look like.`, `An intuitive idea here is as masks increasing, determinism of inverse dynamics would decrease and
our setting would become a multi-modality Imitation Learning, as studied in Shafiullah et al. (2022).`), incorrect use of terminology (eg: `zero-shortly` (change to in a zero short manner), `ill-collected dataset` (change to noisy dataset)) and grammatical errors (eg: `unsupervise RL`) making it challenging to parse the paper.
The introduction of the paper is written in a manner that does not clearly explain the problem, and abruptly switches to listing the contributions. The methods section also lacks clarity as it is difficult to understand sections 3.3 and 3.4. Figure 1 also does not explain the method clearly as the inputs and the outputs of each phase are not shown clearly. Therefore, I would suggest that the authors rewrite certain sections of the paper to make it more understandable.

**Questions:**

- The paper claims that they use `the least amount of human labeling`,  but does not compare it to any other state of the art methods to prove so.
- For training and evaluation purposes, why was the dataset split into 95-5% whereas typically people split it into 70-30% or 80-20%? Also, how was the dataset split to ensure no data leakage between the train and the test set?
- Why are accuracy and precision used as the only metric? Why not report recall and F1 scores as well?
- It is unclear how to interpret Figure 4.

---

### Official Review · Reviewer_xJAs · 2023-11-01

**Soundness:** 2 fair
**Presentation:** 2 fair
**Contribution:** 1 poor
**Rating:** 3
**Confidence:** 4

**Summary:**

This paper introduces Robot-BERT (RoBERT), a method for building robot control policies without human effort. It is trained on datasets collected with unsupervised Reinforcement Learning methods and does not need human labels. It is trained via Masked Action-Inverse-Inference, and is aiming for keyframe-based robot control. Specifically, given the user-supplied state-only keyframe, the robot can achieve the targets with open loop control.

**Strengths:**

Experiments were conducted on three different robots.
Source code and supplementary videos are provided to demonstrate the methods and results.
The approach is based on transformer-based large models.

**Weaknesses:**

The task seems easy to solve. Merely making the robot reach target states does not constitute general control, as this can be achieved with basic control techniques and without jittering observed in the supplementary videos. Moreover, for manipulation tasks using robot arms or dexterous hands, simply reaching target states is insufficient, as interactions with dynamic objects is more important and challenging. If the authors insist on exploring this direction, I recommend considering obstacle avoidance planning for robotic arms or social navigation; and exploring how to encode obstacles into the state to achieve target-reaching while avoiding obstacles.

**Questions:**

Additionally, I recommend not to use the entire field of "ROBOTICS" as a single section in the related work. It would be more effective to focus on specific sub-areas or topics within robotics that are directly relevant to the paper.
Unsupervise RL should be Unsupervised RL.

---

### Official Review · Reviewer_oyMc · 2023-11-01

**Soundness:** 2 fair
**Presentation:** 1 poor
**Contribution:** 2 fair
**Rating:** 3
**Confidence:** 4

**Summary:**

This paper introduces ROBERT which leverages unsupervised data collection and Masked Action-Inverse-Inference (MAII) inspired by Masked Language Modeling to enable robot control without human labels or extensive behavior datasets. The approach demonstrates success in generating stable and flexible robot behaviors across various robot models in simulated environments. The paper also introduces a user-friendly keyframe control mechanism and highlights the multi-task capabilities of the proposed method.

**Strengths:**

- This paper points out an important issue in this field, where existing works rely heavily on extensive amounts of human-labeled multimodal data or expensive experts dataset.
- The research suggests that the fusion of unsupervised RL with masked sequence modeling holds potential for addressing this challenge.

**Weaknesses:**

- The contribution of this work is unclear and the paper lacks novelty, as the proposed method simply combines unsupervised RL dataset with masked modeling without introducing any unique techniques.
- The motivation of adopting a bi-directional sequence modeling approach, akin to BERT-like methods, remains unclear in the context of the problem they are addressing.
- The paper mentions a low-cost aspect without providing details on how this is achieved or its relevance to the main contribution.
- The experiment section of the paper seems to be weak. It is not clear what the experiments aim to show, and there is an absence of any baseline comparisons. Presenting results solely for the proposed method without any comparative baselines seems weird given there are similar line of work such as [1, 2]. It would be better to include experimental results comparing the proposed method to other approaches to show the superiority of the suggested architecture or learning scheme.
- The argument that the proposed method is superior to the state-of-the-art in general robot control is weakened by the fact that their experimental setup focuses on only simple tasks rather than complex tasks (see RT-2[3]). Given the experimental results the authors show, the claim of requiring the least amount of human effort for general robot control/decision-making sounds too strong even though it uses unsupervised dataset.

[1] Philipp Wu et al., Masked Trajectory Models for Prediction, Representation, and Control. ICML, 2023.\
[2] Fangchen Liu et al., Masked Autoencoding for Scalable and Generalizable Decision Making. NeurIPS, 2022.\
[3] Anthony Brohan et al., RT-2: Vision-Language-Action Models Transfer Web Knowledge to Robotic Control. arXiv, 2023.

**Questions:**

- How does the approach of continuously changing the masking ratio randomly during training compare to the CL method? Can additional experimental results be provided to demonstrate this comparison?
- It would be better if the authors add the experiment results that include comprehensive measurements of computational expenses, memory utilization, etc. to show that the proposed method is low-cost.
- As mentioned earlier, it would be beneficial to include experimental results that show the superiority of the proposed approach by comparing it to baselines like MTM[1] and MaskDP[2], all trained with the same unsupervised RL dataset.
- Can complex behaviors, such as interacting with objects, be learned solely from data collected through unsupervised RL? It would be beneficial to include experiments to demonstrate this, as there are significant doubts regarding the capability of learning truly meaningful and practical actions from an unsupervised RL dataset. The claim of this framework being suitable for general robot control remains questionable in terms of its ability to learn valuable actions with real-world relevance solely from an unsupervised RL dataset.
- It is unclear for me about the authors insight about the specified reward function in introduction part. Isn’t it good that the agent has a bias towards the goal when it is learning? Furthermore, as in multi-task RL, if we can specify appropriate reward functions for various tasks clearly to the agent, then the specified reward function itself does not pose a problem for learning multiple behaviors. It is a bit confusing for me regarding the intended argument. Is the main point here that learning multiple behaviors under a single specified reward function is not feasible?

[1] Philipp Wu et al., Masked Trajectory Models for Prediction, Representation, and Control. ICML, 2023.\
[2] Fangchen Liu et al., Masked Autoencoding for Scalable and Generalizable Decision Making. NeurIPS, 2022.